# Adverse pregnancy outcomes and long-term risk of maternal renal disease: a systematic review and meta-analysis protocol

Peter M Barrett,[1,2] Fergus P McCarthy,[2] Karolina Kublickiene,[3] Marie Evans,[3] Sarah Cormican,[4] Conor Judge,[4] Ivan J Perry,[1] Marius Kublickas,[3,5] Peter Stenvinkel,[3] Ali S Khashan[1,2]

¹School of Public Health, University College Cork, Cork, Ireland
²Irish Centre for Fetal and Neonatal Translational Research, University College Cork, Cork, Ireland
³Department of Clinical Sciences Intervention and Technology, Karolinska Institutet, Huddinge, Sweden
⁴Department of Nephrology, Galway University Hospital, Galway, Ireland
⁵Department of Obstetrics & Gynaecology, Karolinska Institutet, Stockholm, Sweden

**Correspondence to**
Dr Peter M Barrett;
peterbarrett1@hotmail.com

## ABSTRACT

**Introduction** Adverse pregnancy outcomes, such as hypertensive disorders of pregnancy (HDP), gestational diabetes (GDM) and preterm birth have been linked to maternal cardiovascular disease in later life. Pre-eclampsia (PE) is associated with an increased risk of postpartum microalbuminuria, but there is no clear consensus on whether HDP increases the risk of maternal chronic kidney disease (CKD) and end-stage kidney disease (ESKD). Similarly, it is uncertain whether GDM, preterm birth and delivery of low birth-weight infants independently predict the risk of maternal renal disease in later life. The aims of this proposed systematic review and meta-analysis are to summarise the available evidence examining the association between adverse outcomes of pregnancy (HDP, GDM, preterm birth, delivery of low birth-weight infant) and later maternal renal disease and to synthesise the results of relevant studies.

**Methods and analysis** A systematic search of PubMed, EMBASE and Web of Science will be undertaken using a detailed prespecified search strategy. Two authors will independently review the titles and abstracts of all studies, perform data extraction and appraise the quality of included studies using a bias classification tool. Original case–control and cohort studies published in English will be considered for inclusion. Primary outcomes of interest will be CKD and ESKD; secondary outcomes will be hospitalisation for renal disease and deaths from renal disease. Meta-analyses will be performed to calculate the overall pooled estimates using the generic inverse variance method. The systematic review will follow the Meta-analyses Of Observational Studies in Epidemiology guidelines.

**Ethics and dissemination** This systematic review and meta-analysis will be based on published data, and thus there is no requirement for ethics approval. The results will be shared through publication in a peer reviewed journal and through presentations at academic conferences.

**PROSPERO registration number** CRD42018110891

## Strengths and limitations of this study

► This proposed systematic review and meta-analysis will provide updated knowledge on the associations between common adverse pregnancy outcomes (ie, hypertensive disorders of pregnancy, gestational diabetes, preterm birth, delivery of low birth-weight infants) and the long-term maternal risk of renal disease.

► The high prevalence of adverse pregnancy outcomes and chronic kidney disease suggests that any potential associations would have important public health implications

► Screening for eligible studies, data extraction and quality assessment will be undertaken by two independent reviewers to minimise the potential for reviewer bias.

► A considerable degree of heterogeneity is anticipated between studies due to differences in diagnostic methods for both exposures and outcomes, and varying lengths of follow-up.

► Only published studies in the English language will be included.

## INTRODUCTION

Chronic kidney disease (CKD) is a major cause of premature morbidity and mortality worldwide, and its prevalence is estimated at 11%–13% among women.[1] End-stage kidney disease (ESKD), although relatively rare, causes disproportionately high healthcare burden and expense.[2 3] Globally, rising levels of obesity, metabolic syndrome and advanced maternal age are resulting in increasing prevalence of adverse pregnancy outcomes, particularly hypertensive disorders of pregnancy (HDP) and gestational diabetes (GDM).[4–6] Pre-eclampsia (PE), gestational hypertension and GDM are now recognised as independent cardiovascular risk factors in women,[7 8] while preterm birth, growth restriction and gestational weight gain have also been linked to future cardiovascular disease.[9–11] By comparison, relatively little is

known about the long-term risk of maternal renal disease following complications of pregnancy.

There is some evidence to suggest that PE increases the risk of maternal kidney disease later in life. A meta-analysis of seven small prospective studies identified an increased risk of microalbuminuria among women who experienced PE compared with those who had normal pregnancies after a mean of 7 years follow-up.[12] A large Norwegian cohort study suggested that pre-eclamptic women were at long-term risk of ESKD.[13] However, the proposed association may be complicated by the strong links between PE and unmeasured cardiovascular risk factors, such as maternal obesity, which can independently increase the risk of CKD.[14]

The long-term risk of CKD among these women has not been comprehensively addressed by studies to date. It is biologically plausible that PE predisposes women to higher risk of CKD; PE is associated with abnormal placental development, and subsequent development of generalised maternal endothelial dysfunction.[15] PE may thus result in lasting renal endothelial damage, which may increase maternal risk of CKD and ESKD.[16] PE is just one of several HDPs, and it is unknown if women who experience gestational hypertension, or PE superimposed on chronic hypertension, are also at increased long-term risk of maternal kidney disease.

A range of other complications of pregnancy may increase the risk of future kidney disease in women. Low birth-weight and preterm delivery have been reported to increase the risk of ESKD in women with PE.[13] However, it is uncertain whether these associations persist independently, or whether birth weight and gestational age are mediators of the association with PE. In one Norwegian cohort study, preterm birth was reported to independently increase the risk of ESKD, but this study was restricted to women with pre-existing diabetes and is not generalisable to the wider population.[17]

GDM is an established risk factor for type 2 diabetes,[18] and may thereby increase the risk of diabetic nephropathy. However, GDM is also an independent risk factor for vascular endothelial dysfunction,[19 20] and is a plausible independent risk factor for maternal kidney disease. To date there has been limited research to examine whether GDM predicts CKD risk independent of subsequent type 2 diabetes.

Worldwide, 3%–5% of pregnancies are complicated by PE[21], 5%–18% of births are preterm[22], 11% have low birth-weight[23] and 5%–13% are complicated by GDM.[24 25] Given the high prevalence of CKD,[1] any true associations between these exposures and CKD may have potentially important public health implications, particularly in resource-poor settings where adverse pregnancy outcomes are more prevalent. Women with complications of pregnancy and at risk of CKD, may benefit from future risk-reduction interventions or enhanced community-based follow-up care to mitigate against possible progression to renal disease. Only a minority of individuals with CKD will ever progress to ESKD, requiring

dialysis or renal transplant, but they experience premature mortality,[26 27] and place a considerable economic and resource burden on health systems.[28]

The aim of this systematic review and meta-analysis is to summarise the available evidence examining the association between adverse pregnancy outcomes and long-term maternal renal disease. The adverse outcomes of interest include HDP (including PE, gestational hypertension, chronic hypertension and PE superimposed on chronic hypertension), preterm birth, delivery of a low birth-weight infant and GDM.

## Population
Women who have had at least one pregnancy of at least 23 weeks gestation.

## Exposures
Any one of the following adverse pregnancy outcomes:
1. Diagnosis of HDP (including PE, gestational hypertension or other HDP).
2. Preterm birth.
3. Diagnosis of GDM.
4. Delivery of a low birth-weight infant (including infants who were small for gestational age).

Any of these adverse outcomes can be defined using established clinical criteria, hospital records or self-reporting of a doctor diagnosis.

## Comparison
Women who never had a corresponding adverse outcome in pregnancy. For example, women who experienced a preterm birth in at least one pregnancy will be compared with women who never experienced a preterm birth.

## Outcomes
Primary outcome 1: CKD.
Primary outcome 2: ESKD.
These primary outcomes can be defined either using established clinical criteria or hospital records.
Secondary outcomes: (1) hospitalisation for renal disease, (2) deaths from renal disease.

## Review question
Does the presence of an adverse pregnancy outcome (ie, HDP, preterm birth, delivery of a low birth-weight infant or GDM) increase the risk of maternal kidney disease in later life?

## METHODS AND DESIGN
This protocol was drafted using the Preferred Reporting Items for Systematic Reviews and Meta-Analysis Protocols checklist.[29] The proposed systematic review and meta-analysis will follow the Meta-analyses Of Observational Studies in Epidemiology (MOOSE) guidelines.[30]

## Search strategy
The lead author (PMB) will undertake a systematic search of the following databases: PubMed, EMBASE and Web

of Science. Peer reviewed journal articles published in the English language, from inception of databases to 31 July 2018, will be included. A detailed search strategy has been compiled, and the search terms will be combined using Boolean Logic where appropriate (AND, OR). The detailed search strategy is available in online appendix 1.

The search of the electronic databases will be supplemented by hand-searching all included papers to identify any further potentially relevant studies.

## Study selection

The titles and abstracts of studies retrieved from each database search will be stored in the EndNote reference manager and de-duplicated. Two review authors (PMB, SC) will screen all titles and abstracts for potentially relevant studies. The full text of the relevant studies will then be retrieved and screened for compliance with eligibility criteria by two reviewers (PMB, CJ). If consensus on eligibility cannot be achieved, a third review author will be consulted (ASK). For any articles which do not meet the inclusion criteria, the reasons for rejection will be noted. A MOOSE flow diagram documenting the process of study selection will be completed.

## Inclusion criteria

► Case–control studies and cohort studies (prospective or retrospective).
► Data provided on an adverse pregnancy outcome of interest (ie, diagnosis of HDP, preterm birth, low birth-weight or GDM) as an exposure variable.
► Data provided on a diagnosis of either CKD or ESKD as an outcome variable.
► Provides a measure of association between one or more of the adverse pregnancy outcomes and CKD or ESKD.
► The diagnosis of maternal CKD or ESKD is made at least 3 months after pregnancy has ended.
► Data must be from an original study.
► Only English-language studies will be considered, including all years from inception of the electronic databases until July 2018.
► Peer reviewed literature only will be included.

## Exclusion criteria

► Non-human studies.
► Studies that are not in English.
► Studies focused on CKD/ESKD risk in the offspring.
► Studies focused on risk of maternal acute kidney injury, without reference to subsequent CKD/ESKD.
► Case reports, case series, letters, commentaries, notes and editorials.
► Studies focused on women with pre-existing renal disease.

## Data extraction

Data from all eligible studies will be extracted by two reviewers (PMB, SC) using a standardised data collection form, including the author and year of publication, the study design, the exposure(s) and outcome(s) of interest,

the definition used for each exposure and outcome, the stage/severity of the outcome (ie, CKD), length of follow-up, the sample size, the confounders adjusted for (if any) and the crude and adjusted measures of association. Where necessary, corresponding authors of published studies will be contacted to obtain any information needed relating to effect estimates. Where effect estimates are not available, absolute numbers of events will be extracted, and crude measures of association will be calculated. If discrepancies arise in data extraction, these will be discussed between reviewers, and where necessary, a third reviewer will be consulted to achieve consensus (ASK).

## Quality appraisal of included studies

The quality of all included studies will be independently assessed by two reviewers (PMB, CJ) using an established quality assessment tool for observational studies. This tool has been described in detail elsewhere.[31] Six types of biases will be assessed: selection, exposure, outcome, analytical, attrition and confounding. For each study, each component will be assigned a risk of bias category: minimal, low, moderate, high or not reported. For example, selection bias will be categorised as 'minimal' if the sample was from a 'consecutive unselected population', whereas selection bias will be categorised as 'high' if sample selection is unclear and if the sample is not representative of the population of interest. For each included study, the overall likelihood of bias will be assessed and reported. Where discrepancies in quality appraisal arise, a third reviewer will be consulted to achieve consensus (ASK).

## Data synthesis and assessment for heterogeneity

Separate meta-analyses will be undertaken for each of the exposure variables and two primary outcomes where possible. Each meta-analysis will be undertaken to calculate the pooled estimate of the relationship between the adverse outcome of interest and subsequent development of maternal CKD or ESKD. For example, for preterm birth as the adverse outcome of interest, a meta-analysis will be undertaken to investigate the association between (1) preterm birth and CKD and (2) preterm birth and ESKD. For HDP, a meta-analysis will be undertaken to investigate the association between (1) any HDP and CKD and (2) any HDP and ESKD. Where possible, subgroup analyses will investigate the associations between PE, gestational hypertension and any other HDP with each of the primary outcomes respectively. For delivery of infants with low birth-weight, a meta-analysis will be undertaken to investigate the association between (1) low birth-weight infant and maternal CKD and (2) low birth-weight infant and maternal ESKD. Where possible, subgroup analyses will investigate the associations between infants small for their gestational age with each of the primary outcomes, respectively.

Both crude and adjusted effect estimates will be displayed using the generic inverse variance method. Adjustment

will be based on the definition outlined in each of the individual studies. Heterogeneity will be explored based on $I^2$ values and $\chi^2$ statistics. Random-effects models will be used if moderate or high levels of heterogeneity are observed between the studies of interest. If studies cannot be meaningfully combined in a meta-analysis, they will be presented in tabular format.

We will perform the following subgroup/sensitivity analyses, where the data allow, using RevMan V.5.3: (1) study type (case–control vs cohort study), (2) stage/severity of CKD, (3) ethnic group, (4) length of follow-up after index pregnancy, (5) number of pregnancies affected by the adverse outcome, (6) measurement of exposure and outcome data (self-reported vs medical records vs laboratory measurements), (7) study quality (minimal/low risk of bias vs moderate/high risk of bias). Furthermore, any studies which have not excluded women with pre-existing renal disease, or have not adjusted for this factor, will be considered in a separate subgroup analysis.

Where 10 or more studies are included in a meta-analysis, we will assess the publication bias. The trim and fill method will be used to identify and correct for funnel plot asymmetry arising from publication bias, if appropriate.[32]

### Ethics and dissemination

This protocol is based on published data, and thus there is no requirement for ethics approval. The results will be disseminated through publication in a peer reviewed journal, and through presentations at academic conferences.

### Patient and public involvement

Patients were not involved in the design of this systematic review and meta-analysis. However, the authors will communicate the study findings to patient and public groups with interest in this area, including action on preeclampsia.

### Potential limitations

There are a number of limitations anticipated in this review. Publication bias may reduce the likelihood of retrieving studies which report non-significant associations between adverse pregnancy outcomes and maternal renal disease. Due to limited resources, only studies which are published in the English language will be included. Publications which only use biomarkers of renal function as outcome variables (eg, microalbuminuria, albumin/creatinine ratio, estimated glomerular filtration rate) will not be included unless they are directly related to a diagnosis of CKD or ESKD.

A degree of heterogeneity is anticipated between studies. Differences in diagnostic methods are likely for both exposures and outcomes. The timeframe for follow-up is likely to vary considerably between studies, with women followed up for longer durations more likely to have developed renal disease. The ability to identify true associations between adverse pregnancy outcomes and future CKD or ESKD will be limited by the length of

follow-up in each of the included studies. Differences in sampling frames are also likely to lead to heterogeneity. Thus, a random-effects model will be used for meta-analyses with moderate or high heterogeneity.

The presence of selection bias and residual confounding is a concern in all observational studies. Potential confounders may include maternal age, ethnic group, socio-economic status, parity, family history, hypertension, diabetes, cardiovascular disease, systemic inflammatory disease, hyperlipidaemia, obesity and smoking. Our meta-analyses will display both crude and adjusted results where possible, basing adjustment on the definition outlined in each individual study. However, given that less adjusted effect estimates may skew the overall results, a sensitivity analysis will be undertaken, where possible, to examine effect estimates which are adjusted more fully for confounders (ie, adjusted for, at a minimum, maternal age, hypertension, diabetes, obesity, smoking, pre-existing kidney disease).

## DISCUSSION

There is a lack of consensus on whether adverse pregnancy outcomes, such as HDP, preterm birth, delivery of a low birth-weight infant and GDM, independently increase the risk of maternal CKD and ESKD. This systematic review and meta-analysis will summarise the available evidence which has examined these associations, thus providing novel information on the role of pregnancy-related factors in the aetiology of maternal renal disease.

**Contributors** PMB, FPMcC, IJP and ASK conceived and designed the protocol, and PMB drafted the protocol manuscript. PMB developed the search strategy, with input from FPMcC, IJP and ASK. PMB, CJ and ASK planned the data extraction. PMB, SC and AK planned the quality appraisal of all included studies. PMB, FPMcC, IJP, SC, CJ, KK, MK, ME, PS and ASK critically revised the manuscript for methodological and intellectual content. All authors approved the final version.

**Funding** This work was performed within the Irish Clinical Academic Training programme, supported by the Wellcome Trust and the Health Research Board (Grant number 203930/B/16/Z), the Health Service Executive National Doctors Training and Planning and the Health and Social Care, Research and Development division, Northern Ireland.

**Competing interests** None declared.

**Patient consent for publication** Not required.

**Provenance and peer review** Not commissioned; externally peer reviewed.

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
