## [Reviewer comments · BMJ Open]

ARTICLE DETAILS

TITLE (PROVISIONAL)	ADVERSE PREGNANCY OUTCOMES AND LONG-TERM RISK OF MATERNAL RENAL DISEASE: A SYSTEMATIC REVIEW AND META-ANALYSIS PROTOCOL.
AUTHORS	Barrett, Peter; McCarthy, Fergus; Kublickiene, Karolina; Evans, Marie; Cormican, Sarah; Judge, Conor; Perry, Ivan; Kublickas, Marius; Stenvinkel, Peter; Khashan, Ali

VERSION 1 - REVIEW

REVIEWER	Elisabetta Versino Department of clinical and biological sciences. University of Torino - Italy I am co author of a similar review, not Yet published and focused specifically to exposure to preeclampsia. Please contact The Principal Investigator, Piccoli Giorgina Barbara. gbpiccoli@ch-lesman.fr
REVIEW RETURNED	19-Nov-2018

GENERAL COMMENTS	We are interested in Sharon our opinioni in this Field, as welln performed a quiter similar review, focused on.preeclampsia
---

REVIEWER	Rachel Kadakia Northwestern University/Ann and Robert H. Lurie Children's Hospital of Chicago
REVIEW RETURNED	29-Dec-2018

GENERAL COMMENTS	The proposed meta analysis will answer the important question of the association adverse pregnancy outcomes on future maternal renal disease. However, the predictability of future disease is going to be limited by the years of follow up in each of the studies that the authors will analyze. 1. The author should provide references for the first sentence of introduction where they state that the stress of pregnancy can predispose to chronic disease2. The introduction wold be stronger if started with a discussion of why renal disease is important- the reader doesn't even know the study is focusing on renal disease until the last sentence of the
--

	first paragraph and the mention of renal disease comes without any context until later in the introduction 3. Please define HDP in the introduction, not just in the abstract 4. How will the authors analyse studies that do not adjust for potential confounders such as GDM or family history of kidney disease etc. This should be more robustly addressed in the limitations section. Results of less rigorously performed studies may skew the results of the meta analysis.
--	---

REVIEWER	Ran Neiger Department of Obstetrics and Gynecology Premier Health Dayton, Ohio, USA
REVIEW RETURNED	13-Jan-2019

GENERAL COMMENTS	The authors may consider adding placental abruption to the list of (relatively) common adverse pregnancy outcomes they are evaluating.
--

VERSION 1 – AUTHOR RESPONSE

Reviewer: 1

Comments received

1. I am co author of a similar review, not Yet published and focused specifically to exposure to preeclampsia.

We are interested in Sharon our opinion in this Field, as welln performed a quiter similar review, focused on.preeclampsia

Response

We note that Reviewer 1 is a co-author on another systematic review which is underway and is registered on PROSPERO. This was brought to our attention by the Deputy Editor of BMJ Open on 28 November 2018, and we provided a response to this on the same date as follows:

“I note a number of differences between this protocol and our own. Firstly, the protocol registered by Piccoli et al. focuses on hypertensive disorders of pregnancy as exposure variables. Our exposure variables include not only hypertensive disorders of pregnancy, but also gestational diabetes, preterm delivery, and delivery of infants who are of low birth weight or small for gestational age. The protocol registered by Piccoli et al. focuses on chronic kidney disease and end-stage kidney disease as outcome variables (similar to our protocol). However, our systematic review will also capture articles which focus specifically on hospitalisation due to renal disease, and mortality due to renal disease as outcome variables. Piccoli et al. plan to search two databases between the years 2000 and 2016, whereas our search strategy will be more comprehensive in its timeframe, including all articles published from inception of three databases (PubMed, Web of Science, EMBASE) to July 2018 inclusive. Furthermore, Piccoli et al. do not plan to appraise the quality of the literature, whereas we have outlined a plan for quality assessment as part of our systematic review.

Thus, while there may be some similarity between these two protocols, our systematic review will provide a considerable amount of additional information, particularly regarding associations between

other (non-hypertensive) exposure variables and long-term renal outcomes. We are confident that our systematic review will contribute a timely, wide-ranging overview of the published literature on adverse pregnancy outcomes and long-term renal outcomes.”

Reviewer: 2

1. The proposed meta analysis will answer the important question of the association adverse pregnancy outcomes on future maternal renal disease. However, the predictability of future disease is going to be limited by the years of follow up in each of the studies that the authors will analyze.

Response

We agree with this observation, and we have added this point to the Limitations section of the protocol. One of the aims of this review is to highlight the gaps in the current literature which will inform the design of future studies on this topic.

2. The author should provide references for the first sentence of introduction where they state that the stress of pregnancy can predispose to chronic disease

Response

Thank you for this suggestion. The first paragraph has now been restructured to address the next comment below, and the sentence relating to the stress of pregnancy has now been removed.

3. The introduction would be stronger if started with a discussion of why renal disease is important- the reader doesn't even know the study is focusing on renal disease until the last sentence of the first paragraph and the mention of renal disease comes without any context until later in the introduction

Response

Thank you for this feedback. We have now restructured the introduction to emphasise the focus on renal disease, and to provide this additional context from the beginning.

4. Please define HDP in the introduction, not just in the abstract

Response

HDP has been spelled out in the first paragraph of the restructured Introduction.

5. How will the authors analyse studies that do not adjust for potential confounders such as GDM or family history of kidney disease etc. This should be more robustly addressed in the limitations section. Results of less rigorously performed studies may skew the results of the meta analysis.

Response

Thank you for highlighting this important point. We have now amended the Limitations section to include a description of our planned sensitivity analysis, which will focus on effect estimates which have been adjusted for a minimum set of important confounders, i.e. maternal age, hypertension, diabetes, obesity, smoking, pre-existing kidney disease.

Reviewer: 3

1. The authors may consider adding placental abruption to the list of (relatively) common adverse pregnancy outcomes they are evaluating.

Response

Many thanks for this suggestion. Although we agree that it may be interesting to consider associations between placental abruption and maternal CKD or ESKD, we feel that this is beyond the scope of the current systematic review. Placental abruption is a rarer complication of pregnancy than the other adverse outcomes listed (i.e. hypertensive disorders of pregnancy, preterm delivery, delivery of low birth-weight or SGA infant, gestational diabetes). Ruiters et al. (2015) conducted a large retrospective cohort study in the Netherlands and reported that 0.2% of mothers experienced a placental abruption. Ananth et al. (2015) also identified declining incidence rates of placental abruption in almost all countries. Furthermore, any identified associations between placental abruption and maternal CKD or ESKD may be confounded by hypertensive disorders of pregnancy, as these factors are strongly linked.